# Phylogenetic Analyses of Some Key Genes Provide Information on Pollinator Attraction in Solanaceae

**DOI:** 10.3390/genes13122278

**Published:** 2022-12-03

**Authors:** Aléxia G. Pereira, Sebastián Guzmán-Rodriguez, Loreta B. Freitas

**Affiliations:** Laboratory of Molecular Evolution, Department of Genetics, Universidade Federal do Rio Grande do Sul, Porto Alegre 91501-970, RS, Brazil

**Keywords:** Solanaceae, molecular evolution, plant-pollinator interaction, ODO1, MYB-FL, NEC1, DFR

## Abstract

Floral syndromes are known by the conserved morphological traits in flowers associated with pollinator attraction, such as corolla shape and color, aroma emission and composition, and rewards, especially the nectar volume and sugar concentration. Here, we employed a phylogenetic approach to investigate sequences of genes enrolled in the biosynthetic pathways responsible for some phenotypes that are attractive to pollinators in Solanaceae genomes. We included genes involved in visible color, UV-light response, scent emission, and nectar production to test the hypothesis that these essential genes have evolved by convergence under pollinator selection. Our results refuted this hypothesis as all four studied genes recovered the species’ phylogenetic relationships, even though some sites were positively selected. We found differences in protein motifs among genera in Solanaceae that were not necessarily associated with the same floral syndrome. Although it has had a crucial role in plant diversification, the plant–pollinator interaction is complex and still needs further investigation, with genes evolving not only under the influence of pollinators, but by the sum of several evolutionary forces along the speciation process in Solanaceae.

## 1. Introduction

The plant–pollinator interaction occurs through a set of floral traits attractive to pollinators that are determined by genes in association with the environment, which favor specific or general interactions. The classic characterization for floral syndromes involves specific interactions that would occur by the convergent evolution of floral traits to the preferences of groups of effective pollinators, which in turn would increase the reproductive success of flowering plants [1,2]. However, this concept is often based on datasets that frequently include qualitative traits, such as flower color, scent, size, symmetry, and orientation, as well as the timing of anthesis, the position of sexual organs, UV-light response, and rewards. Rarely, quantitative characteristics, such as the corolla shape and the length of the corolla tube, are also included. For this reason, there is a recent debate encouraging the exploration of new areas of phenotypic evolution, mainly considering the (post-)genomic era [3].

In Solanaceae, some species have served as models for the identification of genes involved in the transition of floral traits during the pollinator shifts [4,5,6,7,8]. In the genus *Petunia*, the transition from purple to white-flowered species is due to a premature stop codon in the anthocyanin 2 (AN2; an anthocyanin pathway gene) that makes the protein non-functional [9]. Likewise, an extra single-point mutation in the same gene leads to the re-functionalization of a functional protein, which re-introduces the pink-colored flowers [10]. These changes promote the transition from bee pollination observed in short-corolla-tube, pink-colored species, to hawkmoth pollination in long-corolla-tube, white-flowered species and, again, to bee pollination in long-corolla-tube, pink-flowered ones. In *Nicotiana*, the later activation of di-hidroflavonol-4-reductase (DFR; an anthocyanin pathway early gene) during floral development decreases anthocyanidin accumulation, and the interaction between DFR and other genes in the anthocyanin and flavonoid pathways promotes the transition from white to dark-pink flowers and their respective pollinators [11]. The upregulation of the flavonol synthesis pathway due to the changes in the promoter of the transcription factor R2R3 MYB (MYB-FL) promoted the gain of UV absorbance that attracts nocturnal pollinators in *Petunia* species [5], and the loss of this characteristic by the inactivation of MYB-FL was related to the transition to hummingbird pollination [10]. Moreover, changes in the pollinating agent are related to changes in the floral odor emission. The increase in bouquet complexity during the shift from bee to hawkmoth pollination in *Petunia* species involved the modulation of the benzoic acid/salicylic acid methyltransferase (BSMT) and benzoyl-CoA:benzyl alcohol/phenylethanol benzoyltransferase (BPBT) genes in the benzene/phenylpropanoid (FVBP) metabolic pathway, and the loss of function of the cinnamate-CoA ligase (CNL) gene and the reduction in the ODORANT1 (ODO1) expression culminated in the loss of bouquet emission during the shift from the hawkmoth to hummingbird pollination [12].

The analysis of evolutionary changes at DNA and protein levels can allow for the understanding of the evolution of convergent phenotypic traits throughout the floral syndromes and their transitions [13]. The natural selection exerted by the plant–pollinator interactions can act even in convergent adaptive substitutions at diverse taxonomic scales [9,14,15,16]. Phylogenetic relationships among genes rescue the evolutionary history of the sequences, revealing evolutionary pressures, duplications or losses [17], and punctual changes, sometimes producing trees with different topologies when compared with the species trees.

To better understand the evolutionary history of floral syndromes and some genes related to phenotypes considered attractive to pollinators in *Petunia*, we selected the gene DFR from the pigment biosynthesis [18]; the bidirectional sugar transporter (NEC1) which is a crucial component of nectar production in *Petunia* [19]; the MYB-FL, a transcription factor involved in the UV-light response [5]; and the ODO1, which regulates the biosynthesis of floral scent [20]. We obtained the nucleotide sequences of DFR, ODO1, MYB-FL, and NEC1 from available the genomes of Solanaceae species and reconstructed the phylogenetic tree of each gene’s copies. We also tested for the presence of positive selection in each gene. Therefore, we aimed to verify whether genes related to pollinator attraction are conserved among Solanaceae species due to convergence or if their evolutionary history reflects the common ancestry of each genus.

## 2. Materials and Methods

### 2.1. Acquisition and Characterization of Plant Coding Sequences

We conducted a comprehensive homology search based on the BLAST method [21] to obtain the amino acid, genomic, and coding DNA sequences (CDS) of DFR, ODO1, MYB-FL, and NEC1 from the online database National Center for Biotechnology Information (NCBI) (https://www.ncbi.nlm.nih.gov/; accessed on 17 October 2022), and Sol Genomics Network (https://solgenomics.net/; accessed on 17 October 2022). We obtained the *Petunia secreta* genome from NCBI JAFBXY000000000 project PRJNA674325 and the *Petunia exserta* draft genome from DNAZoo (https://www.dnazoo.org/assemblies/Petunia_exserta; accessed on 17 October 2022).

We used the *P. secreta* MYB-FL (KT962949) and *Petunia hybrida* DFR (X79723.1) and ODO1 (AY705977) gene sequences as queries to carry out BLASTn and BLASTp to obtain CDS and proteins, respectively, and *P. hybrida* NEC1 gene (AF313914.1) to carry out tBLASTx and BLASTp for NEC1 CDS and protein sequences. We selected sequences with a cutoff e-value ≥ 2 × 10^−40^ in BLASTn and BLASTp, and e-value = 1 × 10^−70^ in tBLASTn. We selected a new query for each genus when the sequences did not reach such values (Appendix A), improving the sequences’ filtering. All identical, redundant, partial, and incomplete sequences were visually identified and manually eliminated, and only the full-length coding sequences were retained in the final data set.

### 2.2. Multiple Alignments and Gene Structure

To obtain the intron–exon number and size, we aligned the CDSs per gene using the MUSCLE algorithm in MEGA-XI [22] with default settings. When the sequence differed from the expected pattern based on the query genes, we aligned the complete gene sequence and CDS to identify 5′-UTR and 3′-UTR regions and possible issues in genome montage. To check for dubious sequences, we used BLAST in NCBI and removed the doubtful parts if BLAST results matched with a different species than expected or did not match with any species. After that, we evaluated the open reading frame (ORF) in Expasy Web Server (https://web.expasy.org/translate/; accessed on 17 October 2022) using FASTA format and default parameters. We used HMMER (https://www.ebi.ac.uk/Tools/hmmer/search/hmmscan; accessed on 17 October 2022) to find the protein domains for each gene. The secondary structure of such domains was predicted using the amino acid sequences in PSIPRED 4.0 (http://bioinf.cs.ucl.ac.uk/psipred/; accessed on 17 October 2022). The transmembrane regions of NEC1 were predicted using the web TMHMM Server v. 2.0 (https://services.healthtech.dtu.dk/service.php?TMHMM-2.0; accessed on 17 October 2022) with FASTA format and default parameters.

The conserved protein motifs of each gene were investigated in Multiple Em for Motif Elicitation (MEME) v.5.3.3 (http://meme-suite.org/tools/meme; accessed on 17 October 2022) with default parameters, changing the number of motifs until the best sequence coverture was achieved. We generated the multiple protein sequence alignments using GenomeNet Sequence Analysis CLUSTALW (https://www.genome.jp/tools-bin/clustalw; accessed on 17 October 2022) with default parameters. We used these alignments to highlight the conserved amino acid in a BOXSHADE analysis (http://sourceforge.net/projects/boxshade/; accessed on 17 October 2022).

### 2.3. Gene Trees Reconstruction

We reconstructed the phylogenetic tree for each gene using both CDS and protein sequences to explore the evolutionary relationships for DFR, ODO1, MYB-FL, and NEC1 throughout the Solanaceae species. We used *Arabidopsis thaliana* sequences retrieved from The Arabidopsis Information Resource (TAIR; https://www.arabidopsis.org/; accessed on 17 October 2022) as an outgroup. The dataset was filtered by alignment confidence scores (0.003, 99.6% remaining columns) with GUIDANCE [23] using Guidance Web Server (http://guidance.tau.ac.il/; accessed on 17 October 2022). Maximum likelihood phylogenetic analyses were performed using IQ-TREE [24] as implemented in IQ-TREE Web Server (http://iqtree.cibiv.univie.ac.at; accessed on 17 October 2022). We ran 1000 bootstrap (BS) replicates with ultrafast bootstrapping [25] and considered only the branches with BS > 80% as well-supported. The best evolutionary substitution model was set to auto-determination with the ModelFinder option [26] in IQ-TREE. For CDS, the best model based on the Bayesian information criterion (BIC) for each gene was MGK+F1X4_G4 for DFR; GY+F1X4+G4 for ODO1 and NEC1; and TIM3+P+G4 for MYB-FL. For protein sequences, the best BIC-selected models were JTT+G4 for DFR; FLU+G4 for ODO1 and MYB-FL; and CpREV+G4 for NEC1. We used FigTree v.1.4.3 (http://tree.bio.ed.ac.uk/software/figtree/; accessed on 17 October 2022) to visualize the trees and the branch support.

### 2.4. Molecular Evolutionary Analyses

To test for natural selection signatures in genes related to pollinator attraction, we estimated the ratio (ω) between non-synonymous (dN) and synonymous nucleotide substitutions (dS). We used the alignments and respective phylogenetic trees as inputs to investigate natural selection in maximum-likelihood models using CODEML [27] software implemented in EasyCodeML (https://github.com/BioEasy/EasyCodeML; accessed on 17 October 2022). The ω values are a useful measurement for estimating positive selection (advantageous changes if ω > 1), purifying selection (deleterious mutations if ω < 1), or neutral shifts (if ω = 1). We used codon-based models considering variable rates of selection between sites and compared three pairs of site-specific models [28]. The models M3, M2a, and M8 consider the occurrence of positive selection sites (ω > 1), whereas models M0, M1, and M7 are their respective null models. We used the likelihood ratio test (LRT) between the pairs of models to verify which of them better fits our data. We obtained the LRT by calculating twice the log-likelihood difference between pairs of models (2ΔL), with the chi-square (χ^2^) distribution and the number of freedom degrees equal to the number of additional parameters in the more complex model [29]. We considered the ω values not uniformly distributed, and selection could be discussed if the LTR was significant (*p* < 0.01) when comparing M0 vs. M3. A significant LTR could indicate a positive selection when comparing the other model pairs. In these cases, we employed the I Empirical Bayes (NEB) and Bayes Empirical Bayes (BEB) approaches to identify the amino acids that would be under positive selection (posterior probability, PP ≥ 0.9). If the LTR was not significant when comparing M1a vs. M2a, it could indicate that sites were under purifying selection.

## 3. Results

### 3.1. Identification of Gene Sequences in Solanaceae

We obtained the sequences of DFR, ODO1, MYB-FL, and NEC1 genes from 17 complete Solanaceae genomes, *P. hybrida*, and *A. thaliana* (Appendix A). We found a single copy for most genes per species, with a few exceptions (Table 1). The two *Nicotiana attenuata* MYB-FL protein sequences differed by 35.2%; *Nicotiana tomentosiformis* had two MYB-FL identical protein sequences, whereas CDS differed in 25% of sites. The two *A. thaliana* ODO1 proteins differed in 55.6% of amino acids. The two allotetraploid species, *Nicotiana. tabacum* and *Nicotiana benthamiana*, displayed two copies for each gene, except *N. benthamiana* MYB-FL. However, we did not find MYB-FL sequences in *P. hybrida* and *Solanum melongena* or NEC1 in *Capsicum annuum* var. *glabriusculum* and var. *zunla*.

The gene domains identified in HMMER corresponded to those predicted for each gene. For example, all DFR sequences showed a conservative epimerase domain. The NAD(P)H-binding site was highly conserved, except in *A. thaliana*, *N. benthamiana*, and *C. annuum*, the last one with an incomplete sequence. We detected an aspartate (instead of asparagine) at the 143rd site as a conserved site in Solanaceae, except for in *N. benthamiana* (Appendix A). At the 154th site, we found glutamine (instead of the conserved glutamate) in the *Petunia* species.

Two R2R3 domains were observed in most MYB-FL and ODO1 genes (Appendix A). However, only one of these domains was retrieved from ODO1 in *P. inflata* and MYB-FL in *C. annum* varieties. The ODO1 gene had a more conserved R2R3 domain among Solanaceae species than MYB-FL. For the ODO1 R2R3 domain in the *Petunia* species, asparagine at the 27th site and alanine at the 83rd site were replaced by serine residues. In *Capsicum*, alanine at the 19th position was replaced by threonine, and leucine at the 77th site by methionine. In *Solanum* species, we detected the substitution of an aspartate residue by glutamate at the 72nd site. At the 20th site, we observed the conserved glutamate, except in some *Nicotiana* species that had an aspartate in this position. All sequences of NEC1 presented the two MtN3/saliva domain (Appendix A), except *P. secreta* and *C. annuum* var. *zunla*, which had only one domain (Appendix A).

The search for conserved motifs in each gene using MEME revealed that DFR (Figure 1) and ODO1 (Figure 2B) were more conserved among the species than other sequences. For example, in the DFR gene, we identified six motifs (Appendix A) shared among genera, and only *S. lycopersicum* and *S. pimpinellifolium* lost motif 4. Motifs 2 to 6 corresponded to the epimerase domain. In the ODO1 gene, we identified ten motifs (Appendix A); only *P. inflata* lost motifs 4 and 5, and *S. tuberosum* lost motif 5. All ODO1 motifs were observed in all genera, with motifs 1 and 2 corresponding to the R2R3 domain in this gene.

The MYB-FL gene displayed 13 motifs (Figure 2A and Appendix A), 8 of them observed in the *Petunia* species, except for *P. exserta*, in which only 4 motifs were found. Among the *Nicotiana* species, we found seven motifs in *N. tabacum* and *N. tomentosiformis,* which lost motif 8, and 8 motifs in the remaining species, except for in *N. benthamiana*, which had an incomplete gene with only three motifs (motifs 1–3). The *Solanum* species had seven motifs, two of which were exclusive (motifs 9 and 12). No species displayed all the observed 13 MYB-FL motifs. Motifs 1 and 2 corresponded to the R2R3 domain, and only motifs 1 to 3 were shared among all Solanaceae species. *Solanum* and *Capsicum* species shared motif 10, and motif 8 was observed in the *Petunia, Nicotiana*, and *Solanum* species. However, only the *Solanum* species did not differ regarding the number of motifs.

The NEC1 gene had ten motifs (Figure 3 and Appendix A), none of which were exclusive. Motifs 2 and 3 corresponded to the MtN3/saliva domain. The *Petunia* species had nine motifs, all observed in *P. axillaris* and *P. inflata*, and *P. secreta* displayed three motifs, losing motifs 5 to 10. *C. annuum* var. *zunla* had four motifs in the NEC1 gene, and *Solanum* species revealed an inverted order regarding motifs 7, 8, and 10 compared with *Petunia*.

### 3.2. Phylogenetic Relationships

We explored the evolutionary relationships of the DFR, ODO1, MYB-FL, and NEC1 genes through their CDS and protein sequences obtained from Solanaceae species. Sequences from each genus were grouped with high support (Figure 4, Figure 5, Figure 6 and Figure 7), and we observed that all genes more closely reflected the infrageneric evolutionary position than other relationships, with clades corresponding to the *Petunia, Nicotiana, Capsicum*, and *Solanum* genera.

DFR and ODO1 were conserved genes, so the low variation interfered with the positioning of the branches in the phylogenetic tree (Figure 4 and Figure 5). MYB-FL and NEC1 were more variable genes, and the phylogenetic trees showed similar results (Figure 6 and Figure 7). However, the relationships between clades varied depending on the analyzed gene. With that said, the tree’s topology based on these genes was compatible with the phylogenetic signal of the Solanaceae genera.

### 3.3. Analyses of Selection

We estimated the nucleotide substitution rates to detect natural selection acting on genes related to pollinator attraction in Solanaceae (ω = dN/dS). All genes showed significant LTR (*p* < 0.01) values in the comparison of M0 vs. M3, indicating dN > dS (amino acid changing). In M1a vs. M2a, the LTR was not significant (*p* > 0.01) for all genes, indicating purifying selection in ca. 90% of sites in MYB-FL and ODO1, 85% in DFR, and 75% in NEC1, with low divergence during these genes’ evolutions. The M7 vs. M8a of all genes was not significant (Appendix A).

## 4. Discussion

Here, we analyzed four genes that belong to the biosynthetic pathways of compounds involved in the pollinators’ attraction to plants. We characterized these genes from Solanaceae genomes and compared them based on the premise that if the primary driver of diversification in these species was plant–pollinator interaction, then genes should be evolutionarily related following the plant species’ floral syndromes, at least in part, independently from the species relationships.

We selected genes from key pathways related to floral traits, such as visible color, response to UV light, aroma emission, and nectar production. We obtained a scenario for Solanaceae DFR, ODO1, MYB-FL, and NEC1 genes that indicated phylogenetic signals following the species’ evolutionary relationships instead of convergence for these genes. As sampling is essential in phylogenetic studies, the results should be taken with caution [30]. Despite using the genomes deposited in the databases, our sampling for each genus encompassed partial or even reduced proportions (*Petunia*, 4/~15 spp. [31]; *Solanum*, 5/~2000 spp. [32]; *Nicotiana*, 5/~82 spp. [33] and *Capsicum*, 1/~35 spp. [34]). For *Petunia*, which was the main focus of this work, we included the sequences of the two evolutionary clades (long- and short-corolla-tube clades) and the representative species of all floral syndromes in the genus [35].

The scenario obtained in this work reinforces the proposal [36,37] that floral syndromes are more complex and have more fluid processes than initially thought [29], where sets of floral traits associated with particular pollinators’ functional groups are expected to become convergent, and similar floral morphologies would reflect the same kind of pollinator even in distantly related plant taxa [2]. Moreover, pollinators could directionally select the floral traits [38], acting as drivers for diversification [39,40], mainly under conditions of reproductive isolation and adaptation.

Different functional groups mediate pollination in Solanaceae, and several floral syndromes evolved many times in the family. Indeed, there are several examples of species that, despite displaying all traits correlated with a specific syndrome, are pollinated by different functional groups [37].

According to the predicted phenotype for the floral syndromes [1], bee-pollinated species would have visible colors ranging from violet to intense yellow, usually presenting nectar and pollen guides in their petals, producing detectable odor and nectar with a high sugar concentration. As they are pollinated by diurnal insects, these species would not be UV-light-responsive. In turn, hawkmoth-pollinated species would emit a strong nocturnal odor, have white or pale-yellow flowers, respond to UV light, and produce large volumes of diluted nectar. The hummingbird-pollinated species would display flowers, usually with bright colors such as red or orange, without an odor and producing large amounts of nectar with a low sugar concentration.

### 4.1. Visible Colors and UV-Light Response

We analyzed the DFR gene and MYB-FL transcription factor regarding visible colors and UV-light response phenotypes. DFR is part of the anthocyanin biosynthesis pathway that occurs as a single-copy gene in several angiosperm species, such as *Petunia* [41] and *Solanum* [42,43], or duplicated, due to polyploidization as in *Nicotiana* [11].

*N. tabacum* is an allotetraploid and hummingbird-pollinated species that displays flower colors varying from light pink to magenta [44]. Those colors are outside the range of the diploid, white-flowered progenitor *N. sylvestris*, a nocturnal hawkmoth-pollinated species [45], and the dark-pink-flowered *N. tomentosiformis*, a bee-pollinated species [46]. In *N. tabacum*, the reaction between DFR and the product of other genes modulates the flower color [11], with DFR converting the precursors of flavonols into the precursors of anthocyanins. The later activation of DFR during floral development generates high ratios and tends to provide lower anthocyanidin accumulation, resulting in light-pink flowers in *N. tabacum*. The parental species *N. sylvestris* shows low levels of DFR throughout floral development, whereas *N. tomentosiformis* displays higher levels in the initial bud stages [11]. This pattern suggests that DFR and its interaction with other compounds in the anthocyanin pathway are enrolled in the transition between white and dark-pink flowers and their respective pollinators.

The transcription factors anthocyanin 1 (AN1) and anthocyanin 2 (AN2) induce anthocyanin production [18]. The AN2 activates DFR and other genes downstream in the anthocyanin biosynthetic pathway [10]. In *N. tabacum* [47] and *P. axillaris* [9,10], both white-flowered, the AN2 protein is not functional due to a premature stop codon. Moreover, a non-functional AN2 in *Petunia* is considered sufficient to explain its transition from a purple- to a white-flowered species [9], with the regain of a functional gene being responsible for the reversion of the pink color [10]. Similarly, the white flowers in *N. sylvestris* could be related to the absence of an AN2 ortholog in this species [11].

The purple-colored, UV-reflective, and bee-pollinated flowers observed in *P. inflata* represent the ancestral state of the genus *Petunia* [48]. From this broadly distributed ancestor, two divergent lineages derived [49], one with colored flowers (purple corollas and blue pollen), currently represented by the short-corolla-tube species, and another albino lineage that originated the long-corolla-tube species *P. axillaris*, *P. exserta*, and *P. secreta*, which have divergent floral syndromes. *P. axillaris* has white and hawkmoth-pollinated flowers [5,50] and, as in *N. sylvestris*, has entire DFR coding regions. Its capacity to absorb UV, which attracts nocturnal pollinators, occurs with the upregulation of the transcription factor MYB-FL promoter, controlling the expression of genes in the flavonol synthesis pathway [5]. The pink color in *P. secreta,* recovered by the resurrection of the AN2 gene by a single mutation, and the loss of UV absorbance are due to the inactivation of MYB-FL [10].

*P. exserta* has bright red flowers, intensely pigmented with anthocyanins despite an inactive AN2 gene [10]. The DFR of all *Petunia* species rendered non-functional the precursor of the red anthocyanin, dihydrokaempferol [51]. In species that accept dihydrokaempferol, a 26 amino acid region in DFR determines the substrate specificity, with a conserved asparagine at the 134th site and glutamate at the 145th site [52] which correspond to the 143rd and 154th positions in our alignment, respectively. In *Petunia*, these amino acids are replaced with aspartate (the most common among the Solanaceae species) and glutamine, respectively. The molecular specificity mechanism is unknown, but this can alter the substrate recognition by DFR [52]. In addition, *P. exserta* loses the UV absorbance ability due to a frameshift mutation in MYB-FL that inactivates a flavonol pathway [5]. UV-reflectance is a relevant characteristic of red flowers [53], such as *P. exserta,* that attracts hummingbirds [54]. Regaining red anthocyanins involves the upregulation of other MYB transcription factors that replace the ancestral function of AN2 [55]. Similar to *P. exserta, N. benthamiana* has an incomplete MYB-FL protein and, unexpectedly, due to its polyploid condition, only one gene copy was recovered from the genome. It is inferred from these characteristics that *N. benthamiana* could be UV-light-reflective, but further investigation is still necessary.

The DFR gene shows well-conserved sequences, and we recovered the entire coding region, even from species with white flowers, which do not produce anthocyanins. However, the losses and shifts in flower pigmentation are related to changes in gene expression, a different phase of development, or different color pathways and not necessarily to structural mutations in this gene. For example, the MYB-FL was more diverse, with mutations that impacted the protein’s function. Different colors and UV-light responses enable pollinators to identify rewards associated with their needs and preferences. Vibrantly colored flowers, visual tracking, and UV-reflectance are associated with diurnal pollinator attraction, affecting species such as bees and hummingbirds [56], whereas white and UV-absorbent flowers are usually related to nocturnal pollinators, such as hawkmoths [57].

### 4.2. Odor Emission

The ODO1 is a transcription factor that regulates the benzene/phenylpropanoid (FVBP) metabolic pathway responsible for the biosynthesis of floral scents [58]. In *P. axillaris* and *P. hybrida* var. Mitchell, benzenes are the main volatiles emitted and exert strong levels of attraction for the hawkmoth *Manduca sexta* [9]. These volatiles are primarily produced in petals during the night, coinciding with the peak foraging activity of the pollinator [59]. The quantity and complexity of the volatiles emitted by *P. axillaris* and *P. hybrida* are related to the expression of genes in the benzoic acid pathway. Such genes moderately increase the expression of some proteins in the FVBP [12]. The scent absence in *P. exserta* is related to the inactivation of the cinnamate-CoA ligase (CNL) gene and not to the loss of function in the ODO1 [60]. In turn, the bee-pollinated *P. inflata* emits volatile benzaldehyde, but no other compounds are present in *P. axillaris* [30]. This difference in scent composition and emission is related to the differential expression of genes in the FVBP pathway. The absence of methyl benzoate in *P. inflata* aroma is attributed to a blockage in the last step in the pathway [12].

Similar to what was found in the *Petunia* clade, the *Nicotiana* species exhibit great diversity in terms of pollinators. Moth- and hawkmoth-pollinated species such as *N. sylvestris* and *N. attenuata* emit scents at night, regulated by the circadian cycle [61]. The predominant floral fragrance is benzyl alcohol in *N. sylvestris* [62] and benzyl acetone in *N. attenuata* [63]. Moreover, *N. attenuata* has a small number of flowers that open at dusk, interacting with day-active pollinators that are still foraging at that time, and there are suggestions that, despite hummingbirds not being guided by scent, benzyl acetone may work as an attractant in the taste of nectar [63].

### 4.3. Floral Rewards

Many plants use nectar as a floral reward, and its composition meets the preferences of pollinators [64]. NEC1 is expressed in floral nectaries, acting as a sugar efflux transporter, which contributes to the sugar concentration and composition of nectar [18]. The *Nicotiana* bee-pollinated or autogamous species show a reduced volume of nectar; hummingbird-pollinated species have a slightly higher volume of nectar; and the hawkmoth-pollinated species produce large volumes of nectar [65]. These patterns were also detected in *Petunia* species with the same floral syndromes [6,64,66].

Pollinated by bees, *P. inflata* and *C. annuum* have low volumes of nectar, consisting predominantly of glucose and fructose. In *P. inflata*, the sucrose concentration is proportional to the other sugars [64,66]. *P. secreta* has nectar with a concentration of sugars similar to that observed in hummingbird-pollinated species and a higher volume of nectar than expected for bee-pollinated species. However, due to the long floral tube, bees cannot access the nectar [36]. The NEC1 gene in *P. secreta* was recovered as an incomplete sequence, although sugar is found in the nectar, which requires further research on the gene functionality and sugar synthesis pathway. The *Solanum* species have nectar-less flowers, and pollen is the floral reward for pollinators [67], similar to in *P. secreta* [36].

### 4.4. Gene Conservation by Purifying Selection

Our results indicated that DFR, MYB-FL, ODO1, and NEC1 are under purifying selection. Flavonols are essential not only for floral attraction but also for controlling pollen-tube growth and high-temperature stress [68], with MYB-FL being one of the first regulators in this pathway [10]. DFR is conserved among white-colored species [55]. However, anthocyanins are also present in vegetative organs and fruits, protecting against the effects of UV light and temperature stresses [69,70,71]. In odorless species such as *P. exserta*, pathway inactivation does not occur by the modification of ODO1 [60], and NEC1, in addition to the transportation of sugar to nectar, has also been linked to the development of anthers and sugar transportation in fruits [72]. The conservation of the sequence and the maintenance of function by purifying selection indicates the importance of these genes for different pathways, not necessarily linked to pollinator attraction.

## 5. Conclusions

Our results revealed that the genes recovered the infrageneric phylogenetic relationships, with clades corresponding to the studied genera. Many genes reflected the species’ most recent common ancestor within each genus. We found that DFR and ODO1 were the most conserved, whose phenotype shifts were not related to mutations but more likely to a differential expression among Solanaceae species. The inactivating mutation in MYB-FL seems to be the key to pollinator shifts, despite its well-conserved function in most Solanaceae species. The analyzed genes are related to the phenotypes attractive to pollinators, but their products and biosynthetic pathways also play other vital roles in plants. The history of these genes was probably designed not only by the interaction with pollinators but by the sum of several evolutionary forces along the speciation process in Solanaceae.

## Figures and Tables

**Figure 1 genes-13-02278-f001:**
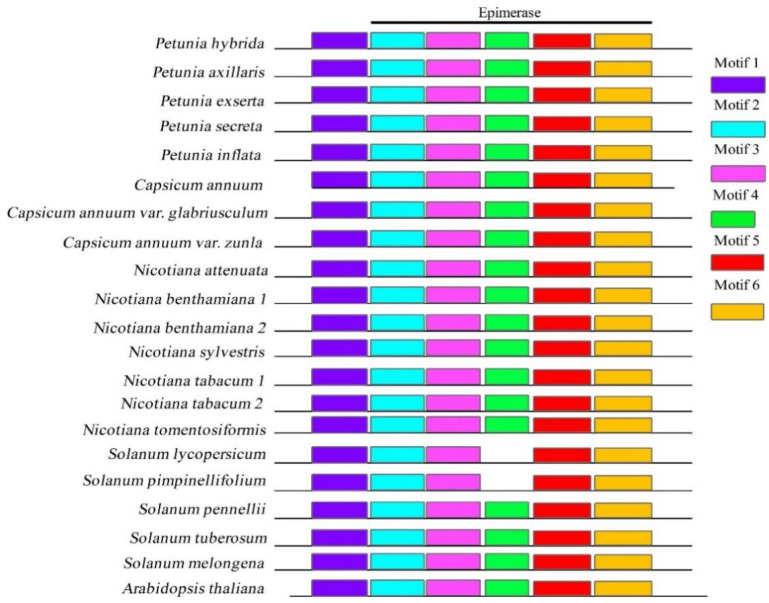
Conserved motifs of DFR protein sequences. The motifs are indicated by a numbered box and different colors; gray lines indicate non-conserved sequences. The motifs’ length is proportional to the sequence size. The epimerase domain is indicated on top by a black line.

**Figure 2 genes-13-02278-f002:**
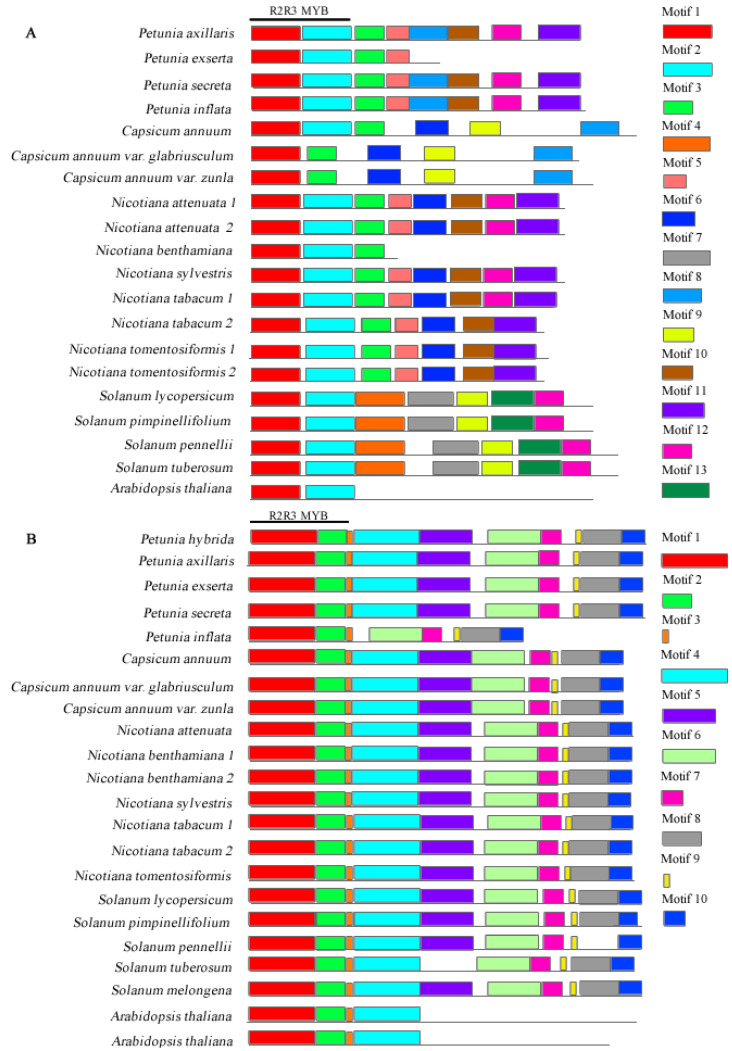
Conserved motifs of (**A**) MYB-FL and (**B**) ODO1 protein sequences. Both genes are part of the MYB family. The motifs are indicated by a numbered box and different colors; grey lines indicate non-conserved sequences. The motifs’ length is proportional to the sequence size. The R2R3 domain is indicated on top by a black line.

**Figure 3 genes-13-02278-f003:**
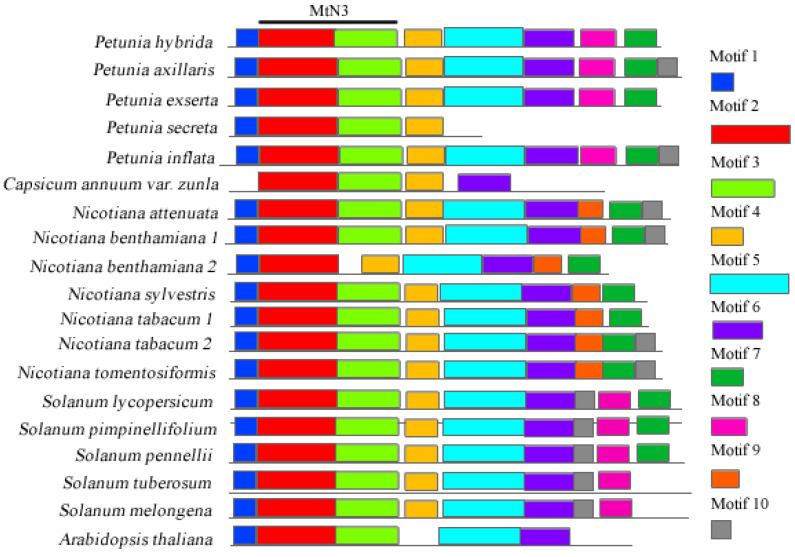
Conserved motifs of NEC1 protein sequences. The motifs are indicated by a numbered box and different colors; grey lines indicate non-conserved sequences. The motifs’ length is proportional to the sequence size. The MtN3 domain is indicated on top by a black line.

**Figure 4 genes-13-02278-f004:**
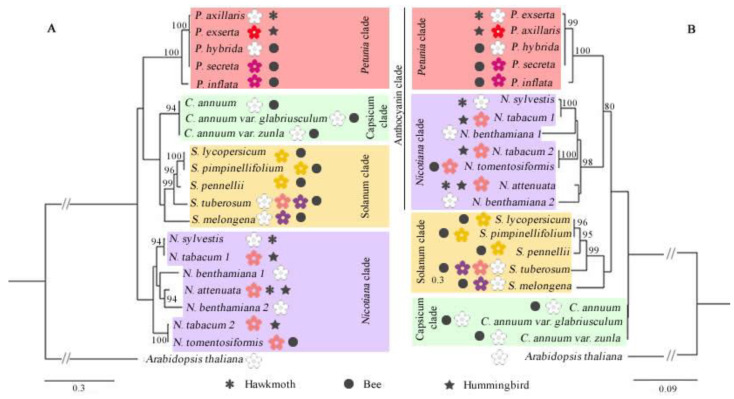
Phylogenetic relationships of DFR gene sequences retrieved from 19 complete Solanaceae species genomes. Maximum likelihood phylogenetic tree based on CDS sequences (**A**) and protein sequences (**B**). Only bootstraps > 80% are shown above the branches. The dark symbols indicate the pollination syndrome and colored flower cartoons represent each species’ visible flower color. The absence of pollination syndrome symbol in *N. benthamiana* indicates autogamy in this species.

**Figure 5 genes-13-02278-f005:**
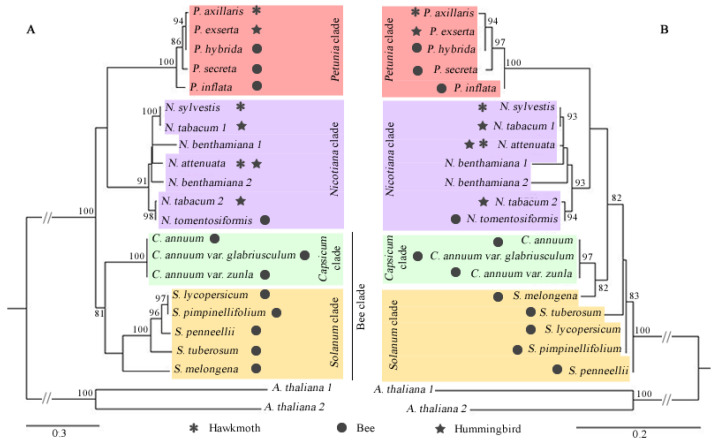
Phylogenetic relationships of ODO1 gene sequences retrieved from 19 complete Solanaceae species genomes. Maximum likelihood phylogenetic tree based on CDS sequences (**A**) and protein sequences (**B**). Only bootstrap > 80% are shown above the branches. The dark symbols indicate the pollination syndrome and the absence of pollination syndrome symbol in *N. benthamiana* indicates autogamy in this species.

**Figure 6 genes-13-02278-f006:**
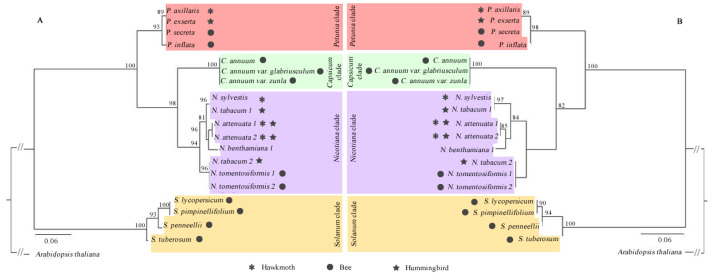
Phylogenetic relationships of MYB-FL gene sequences retrieved from 19 complete Solanaceae species genomes. Maximum likelihood phylogenetic tree based on CDS sequences (**A**) and protein sequences (**B**). Only bootstraps > 80% are shown above the branches. The dark symbols indicate the pollination syndrome and the absence of pollination syndrome symbol in *N. benthamiana* indicates autogamy in this species.

**Figure 7 genes-13-02278-f007:**
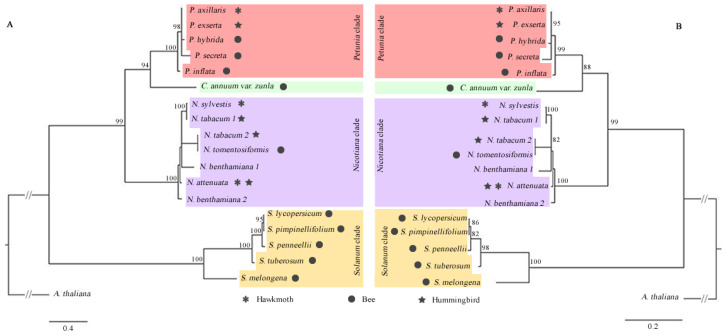
Phylogenetic relationships of NEC1 gene sequences retrieved from 19 complete Solanaceae species genomes. Maximum likelihood phylogenetic tree based on CDS sequences (**A**) and protein sequences (**B**). Only bootstraps > 80% are shown above the branches. The dark symbols indicate the pollination syndrome and the absence of pollination syndrome symbol in *N. benthamiana* indicates autogamy-pollination in this species.

**Table 1 genes-13-02278-t001:** Solanaceae species with complete genomes available at Sol Genomics and NCBI, highlighting the species ploidy and number of sequences recovered per species for the genes di-hydroflavonol-4-reductase (DFR) and nectar 1 (NEC1), and the transcription factors odorant 1 (ODO1) and MYB-FL.

Species	Ploidy	Copy Number				Pollinator
		DFR	ODO1	MYB-FL	NEC1	
*Petunia axillaris*	2n	1	1	1	1	Ha
*Petunia exserta*	2n	1	1	1	1	Hu
*Petunia hybrida*	2n	1	1	0	1	B
*Petunia inflata*	2n	1	1	1	1	B
*Petunia secreta*	2n	1	1	1	1	B
*Capsicum annum*	2n	1	1	1	1	B
*Capsicum annum* var. *glabriusculum*	2n	1	1	1	0	B
*Capsicum annuum* var. *zunla*	2n	1	1	1	0	B
*Nicotiana attenuata*	2n	1	1	2	1	Ha/Hu
*Nicotiana benthamiana*	4n	2	2	1	2	X
*Nicotiana sylvestris*	2n	1	1	1	1	Ha
*Nicotiana tabacum*	4n	2	2	2	2	Hu
*Nicotiana tomentosiformis*	2n	1	1	2	1	B
*Solanum lycopersicum*	2n	1	1	1	1	B
*Solanum melogena*	2n	1	1	0	1	B
*Solanum pennellii*	2n	1	1	1	1	B
*Solanum pimpinellifolium*	2n	1	1	1	1	B
*Solanum tuberosum*	2n	1	1	1	1	B
*Arabidopsis thaliana*	2n	1	2	1	1	-
Total		21	22	20	19	

2n—diploid; 4n—allotetraploid; Ha—hawkmoth; Hu—hummingbird; B—bee; X—*N. benthamiana* is autogamous.

## Data Availability

Not applicable.

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
