# Peer review of "Phylogenetic Analyses of Some Key Genes Provide Information on Pollinator Attraction in Solanaceae"

_genes, 2022, doi:10.3390/genes13122278_

Round 1

Reviewer 1 Report

In this paper, authors have used a phylogenetic approach to understand genes involved in biosynthetic pathways responsible for some phenotypes attractive to pollinators in Solanaceae. It’s a well-written article, especially the results and discussion section. I have a few issues with the introduction. It lacks a literature review, and there are many scopes to make it more engaging.

My comments

Abstract:

L9-L10 “Floral syndromes are known by the conserved morphological traits in flowers associated 9 with pollinator attraction” In terms of pollination syndrome not all traits are conserved, it might be good to rewrite this sentence.

L15-L16: Not clear

Introduction:

L26  How about flowering  plants?

L28- It can also be similar functional groups

L 30-31 Needs more elaboration

L33-36 onwards..  The central objective is straight forward and author requires focusing on the questions and building the introduction around it. Introduction can be made little longer and before reaching the hypothesis/question author should be able to convince the reader why asking these questions are important.  Clarity and link between each sentence can be improved. Make sure that each paragraph has a larger message and the sentences are all tied up well and contribute to your larger message. Authors need to do more literature review and cite papers where genes related to pollinator attraction are studied to understand the pollination syndrome in other plant systems.  Pollination syndrome is a well studied topic and it is easy to find papers in support of your hypothesis and opposite to it. Both needs to be discussed. Terms like phylogenetic signal need to be introduced in the introduction itself because it’s the central part of this paper.

Author Response

Comments and Suggestions for Authors

In this paper, authors have used a phylogenetic approach to understand genes involved in biosynthetic pathways responsible for some phenotypes attractive to pollinators in Solanaceae. It’s a well-written article, especially the results and discussion section. I have a few issues with the introduction. It lacks a literature review, and there are many scopes to make it more engaging.

My comments

Abstract:

L9-L10 In terms of pollination syndrome not all traits are conserved, it might be good to rewrite this sentence.

 “Floral syndromes are known by the conserved morphological traits in flowers associated with pollinator attraction”

A: We rephrased this sentence.

L15-L16: Not clear

A: We rephrased this paragraph to better explain our meaning

Introduction:

L26 How about flowering plants?

L28 - It can also be similar functional groups.

L 30-31 Needs more elaboration.

L33-36 onwards. The central objective is straight forward and author requires focusing on the questions and building the introduction around it. Introduction can be made little longer and before reaching the hypothesis/question author should be able to convince the reader why asking these questions are important.  Clarity and link between each sentence can be improved. Make sure that each paragraph has a larger message and the sentences are all tied up well and contribute to your larger message. Authors need to do more literature review and cite papers where genes related to pollinator attraction are studied to understand the pollination syndrome in other plant systems.  Pollination syndrome is a well studied topic and it is easy to find papers in support of your hypothesis and opposite to it. Both needs to be discussed. Terms like phylogenetic signal need to be introduced in the introduction itself because it’s the central part of this paper.

A: We modified the Introduction section, including  more examples and details that justify our aims.

Reviewer 2 Report

12 November 2022

Dear Editor/Associate Editor,

Article: Phylogenetic analyses of some key genes provide information on pollinator attraction in Solanaceae

Due: 14 November 2022

The authors’ attempt to unveil the effect of specific floral development genes on pollination syndromes in Solanaceae seems to be courageous. I can see that the study is prone to several limitations which make accepting this article a questionable task. The authors made use of the vast sequencing data available for the megadiverse, economically important Solanaceae, for inferring the eco-evolutionary interactions in the group. The inferences do not yet convince me of what the authors put forward, as these complex eco-evolutionary interactions are not shaped solely by the loss or gain of genetic structure and function. Instead, a vast array of epigenetic and ecological factors might have attributed to the shaping of patterns we see in nature today. Yet, I appreciate the authors for

Concerning the current manuscript, I have a few questions/suggestions that are listed here:

1.       Although the methods and results sections of the manuscript are comparatively well-written, critical information is missing, and those details are highlighted in the attached file. As sampling constitutes the crux of any phylogenetic study, I am also curious if the inferences would remain the same when missing lineages are added to the existing tree. As far as I can see that, the proportion of sampling for each of the studied genera is as follows: Petunia (4/~17 spp.), Solanum (5/~1289 spp.), Nicotiana (5/~89 spp.), Capsicum (1/~42 spp.), and Arabidopsis (1/~11 spp.; POWO, 2022)  It would be great if the authors could mention this as a caveat of their current study in the discussion section. The restricted sampling can result in wrong inferences (Craig et al. 2022) such as flawed hypotheses regarding the evolution of traits. Hence, the authors should provide the proportion of sampling for each genus in terms of infrageneric representation. The focal taxa can easily be identified for future sampling and genome sequencing.

2.       Further, coherence is missing from the introduction, and the authors could make it more impactful by being more insightful. I am not too impressed by the first paragraph in its current state. This is also true for the later paragraphs, and they can be improved after providing more information on studies that relied on similar approaches. At the current state, the introduction is too focused on the Petunia clade, and this will not help in reaching a broad audience. The authors can enlighten the audience by bringing the pollination strategies in other Solanaceae. This will also help them shape their discussion more broadly and make it interesting to a diverse audience. Further, in the discussion section, the authors highlight the gene functions, which anyway have no connection with whatsoever they discussed in the other sections.

3.       More importantly, a justification for the choice of their phylogenetic analyses is not well provided. I am curious to see the congruence among topologies obtained from different frameworks (Bayesian, Parsimony, Likelihood, and other coalescent-based approaches). If the authors had thought about it, they should include it in the discussion section. If not, then a justification is warranted. Also, a phylogenetic analysis of the combined (all four genes) would have been interesting given it would provide a better resolution. What do you think?

4.       If possible, the authors should also analyze to determine the phylogenetic signal for each of the floral/pollination syndromes identified. This will help them understand the transitions in character-states across taxa and can also be read as evidence in response to the changes in genotype or phenotype.

5.       A dataset is missing for the pollination syndromes, as currently, the criteria for coding pollinator interactions are not provided alongside the manuscript.

6.       My major concern, again, is the limitations of this dataset- as how prone this would be to adding more data.

Suppose the authors can consider incorporating these changes. In that case, I believe the manuscript would have the potential to cater to a broad audience, due to the immense ecological and evolutionary importance of Solanaceae. At the current state, I am not so happy with the overall content of this paper, as more analyses can add more value to this study.

Some of my comments are included in the attached file.

Thank you abundantly for this opportunity, and I wish the authors a great review experience.

Cheers,

Author Response

Comments and Suggestions for Authors

The authors’ attempt to unveil the effect of specific floral development genes on pollination syndromes in Solanaceae seems to be courageous. I can see that the study is prone to several limitations which make accepting this article a questionable task. The authors made use of the vast sequencing data available for the megadiverse, economically important Solanaceae, for inferring the eco-evolutionary interactions in the group. The inferences do not yet convince me of what the authors put forward, as these complex eco-evolutionary interactions are not shaped solely by the loss or gain of genetic structure and function. Instead, a vast array of epigenetic and ecological factors might have attributed to the shaping of patterns we see in nature today. Yet, I appreciate the authors for concerning the current manuscript, I have a few questions/suggestions that are listed here:

  1. Although the methods and results sections of the manuscript are comparatively well-written, critical information is missing, and those details are highlighted in the attached file. As sampling constitutes the crux of any phylogenetic study, I am also curious if the inferences would remain the same when missing lineages are added to the existing tree. As far as I can see that, the proportion of sampling for each of the studied genera is as follows: Petunia(4/~17 spp.), Solanum (5/~1289 spp.), Nicotiana (5/~89 spp.), Capsicum (1/~42 spp.), and Arabidopsis (1/~11 spp.; POWO, 2022)  It would be great if the authors could mention this as a caveat of their current study in the discussion section. The restricted sampling can result in wrong inferences (Craig et al. 2022) such as flawed hypotheses regarding the evolution of traits. Hence, the authors should provide the proportion of sampling for each genus in terms of infrageneric representation. The focal taxa can easily be identified for future sampling and genome sequencing.

A: We add this information in the Discussion and also included more details that justify our aim.

  1. Further, coherence is missing from the introduction, and the authors could make it more impactful by being more insightful. I am not too impressed by the first paragraph in its current state. This is also true for the later paragraphs, and they can be improved after providing more information on studies that relied on similar approaches. At the current state, the introduction is too focused on the Petuniaclade, and this will not help in reaching a broad audience. The authors can enlighten the audience by bringing the pollination strategies in other Solanaceae. This will also help them shape their discussion more broadly and make it interesting to a diverse audience. Further, in the discussion section, the authors highlight the gene functions, which anyway have no connection with whatsoever they discussed in the other sections.

A: We modified the section Introduction to attend reviewers' suggestions and queries.

  1. More importantly, a justification for the choice of their phylogenetic analyses is not well provided. I am curious to see the congruence among topologies obtained from different frameworks (Bayesian, Parsimony, Likelihood, and other coalescent-based approaches). If the authors had thought about it, they should include it in the discussion section. If not, then a justification is warranted. Also, a phylogenetic analysis of the combined (all four genes) would have been interesting given it would provide a better resolution. What do you think?

A: We used a classical methodology to study gene and protein evolution that contemplates birth-death evolutionary model and adequate substitution models. Most works with similar goals use the same. The genera phylogenies were compared with genes' trees only to contraste genes position with species position. The analysis based on concatenated genes was not included because the large missing data due to differences in recovered sequences number.

  1. If possible, the authors should also analyze to determine the phylogenetic signal for each of the floral/pollination syndromes identified. This will help them understand the transitions in character-states across taxa and can also be read as evidence in response to the changes in genotype or phenotype.

A: We included the floral syndrome for each analysed species based on previously published works. We lack the information on floral syndromes evolution for most the included genera. Only in Petunia (our group results, previously published elsewhere), we know that from a bee-pollinated ancestor (like P. inflata), a white-flowered hawkmoth derived lineage appeared which gave arise to a red-flowered hummingbird-pollinated and a pink coloured bee pollinated species.

  1. A dataset is missing for the pollination syndromes, as currently, the criteria for coding pollinator interactions are not provided alongside the manuscript.

A: We included the known information in the figures.

  1. My major concern, again, is the limitations of this dataset- as how prone this would be to adding more data.

A: Reviewer is totally right, new genomic information for other Solanaceae species may (or not!) change completely our results on the genera, except Petunia that was our main focus. For this genus we included representative of the two clades and all floral syndromes and we think our results are strong and well-supported. In the future, we might use a different strategy, focus in other questions, and compare the new results with this work.

  1. In the manuscript both pollination and floral syndromes are used. Isn’t better use only one term?

A: We changed all for "floral syndromes".

  1. “The plant-pollinator interaction is complex and still needs more investigation, although has had a crucial role in plant diversification.” A much more clear conclusion is expected.

A: We changed this paragraph to better explain our meaning.

  1. The figure 1 need to be corrected.

A: Corrected.

  1. L 290 – 292 “We characterized these genes from Solanaceae genomes and compared them based on the premise that if the primary driver of diversification in these species was plant-pollinator interaction, then genes should be evolutionarily related following the plant species’ floral syndromes.” I can’t agree with this statement. The ecological factors like pollinator attraction often cause divergence than convergence in the phenotypes among plants. So how can you say that these are evolutionary related?

A: In this paragraph we described our hypothesis for genes evolution under convergence. The reviewer is completely right to say that plant-pollinator interaction can simultaneously produce divergence and convergence depending the comparison.  In this work, we supposed that pollinators would be the selective pressure, so, if it was true, we would expect that, for example, P. inflata and P. secreta would be in the same clade in all trees, because both attract the same group of pollinators (bees).

Reviewer 3 Report

1-Remove the "e.g" word in the number of references in the text.

2-Add the information in the citation section.
3- Improve the English language
4-Clear explain and shorten the sentences in the material and method section.

Author Response

Comments and Suggestions for Authors

1-Remove the "e.g" word in the number of references in the text.

A: done

2- Add the information in the citation section.

A: done

3- Improve the English language

A: manuscript was reviewed by MDPI English service (certificate attached)

4-Clear explain and shorten the sentences in the material and method section.

A: We rephrased several sentences accordingly.

Round 2

Reviewer 3 Report

This article could be accepted.